

# Output voltage control of DAB converters based on uncertainty and disturbance estimation

Haijun Tian,Zheng Zhou,Yuanshuai Liu,Yuepeng Zhang and Mingsuo Yang

School of Automation Engineering, Northeast Electric Power University, Jilin, Jilin, China

## ABSTRACT

The dual active bridge (DAB) converter is a power electronic device commonly used for DC voltage regulation and stabilization. However, during its control process, external disturbances, load variations, input voltage variations, switching tube voltage drops, dead time, *etc*. lead to errors in the control output, thus reducing the control accuracy of the system. Therefore, this article propose a robust control scheme for the output voltage based on uncertainty and disturbance estimator. In this article, an average small-signal model of the dual active bridge converter was established in terms of the basic principles and operation mechanisms, simplifying the controller's design. Then, the basic principles of the uncertainty and disturbance estimator (UDE) are introduced. The small-signal model of the dual active bridge (DAB) converter is applied to the UDE to minimize output voltage error by enabling the controller to directly regulate the shift ratio. Finally, this article discusses the application and effectiveness of the uncertainty and disturbance estimator (UDE) in the simulation and control of dual active bridge (DAB) converters. A series of experimental comparative studies are conducted, demonstrating that this scheme offers significant advantages in suppressing system uncertainties and disturbances.

# INTRODUCTION

Initially, the dual active bridge converter was proposed by *De Doncker & Divan (1991)*. After more than 30 years of development, it has been widely used due to its benefits, including bi-directional power transfer, a wide voltage regulation range, and zero-voltage turn-off (*Shao et al., 2019*). However, during the control of a dual active bridge converter, the external perturbations, load variations, input voltage fluctuations, switching tube voltage drops, dead time, and other factors lead to errors in the control output, thus reducing the control accuracy of the system. Therefore, various control methods have been proposed for output voltage control.The simplest method to regulate the output voltage feedback is through a proportional-integral (PI) controller, which adjusts the phase-shift ratio based on output voltage feedback. Additional feed-forward paths can be incorporated to enhance the PI controller and further improve output voltage regulation (*Qin & Kimball, 2014*). The linearization control method also belongs to a kind of PI control, which uses linearization

Corresponding author
Haijun Tian, 20101305@neepu.edu.cn

to eliminate the nonlinear term in the DAB, and this control can reduce the sensitivity of the system stability to the load condition and the reference voltage, and enlarge the stability margin (*Tong et al., 2019*). While the feed-forward compensation strategy indeed improves the transient response of the DAB to output load variations, its effectiveness comes at the cost of increased computational complexity (*Segaran, Holmes & McGrath, 2013*). Another drawback is the swift decline in control performance when system parameter uncertainty is present. The disturbance observer feed-forward compensates the observed ensemble disturbance at the control input. This collective disturbance encompasses the system's uncertain discordant disturbances. Therefore, disturbance-observer based control (DOBC) can attain superior control performance (*Ali et al., 2019*). Predictive control stands as an alternative method employed for output voltage regulation, differing from feedback control. In feedback control, parameter selection predominantly relies on trial and error or machine learning, wherein improper parameter choices may induce system instability. Consequently, the design of subparameters becomes considerably intricate (*Chen et al., 2020*; *Chen et al., 2019b*). Another control method widely used in DAB systems is sliding mode control(SMC), the traditional SMC has the disadvantage of jitter, which becomes a serious problem in the control of the DAB converter, so the research on the problem focuses on how to reduce the system jitter vibration, Such as double integral sliding mode control (*Jeung, Choi & Lee, 2016*), super-twisted sliding mode control (*Tiwary et al., 2023*), and so on.

In addition, for the uncertainty and disturbance of the system, the method of uncertainty and disturbance estimator(UDE) is a well-established method, which was proposed by Qing-Chang Zhong in 2004 (*Zhong & Rees, 2004*), and then successively applied in nonlinear (*Deshpande & Phadke, 2012*), discrete (*Padmanabhan, 2021*), and non-ray-imitating (*Ren, Zhong & Chen, 2015*) systems, such as in permanent magnet synchronous motors (*Ren et al., 2017*), dynamic positioning of vessels (*Huang et al., 2021*), and LCL-type grid-tied inverters (*Xiong, Ye & Zhu, 2023*). For DC-DC converters, particularly in the context of regulating lift-voltage converter output voltages, this method has proven successful. Here, the controller ensures nominal performance across the full operating range by swiftly assessing and mitigating uncertainties and disturbances (*Tian et al., 2019*). *Wu et al. (2019)* present an output power model for DAB converters, amalgamating a DAB circuit model with a phase shift scheme to offer a comprehensive model for DAB converters. Based on this model, a UDE-based voltage controller is designed and the effectiveness of the controller is verified. *Wu et al. (2020)* applies the UDE control method for a constant power load of a DAB converter. for the first time, the output impedance of a UDE-controlled DAB converter is modeled and analyzed for stability. So far, the control of DAB converter mostly starts from its power, and few directly control its the shift ratio to realize the stable output voltage.

In terms of system modeling for DAB, there are mainly average models (*Rodríguez Alonso et al., 2010*; *Chen et al., 2019a*), generalized average models (*Qin & Kimball, 2012*), and discrete time models (*Shi et al., 2017*). The averaging model is a method that ignores the dynamics of the inductor current and only considers the voltage dynamics at the output. The generalized average model takes into account the inductor current as well

as the voltage at the output, and it uses the Fourier transform to handle time-varying and periodic physical quantities. The generalized averaging model necessitates a balance between accuracy and complexity. Both aspects escalate with the inclusion of additional Fourier series terms in the model. Similar to the generalized averaging approach, the discrete-time model considers the dynamics of the current, treating the state variables as evolving at distinct time intervals, thus introducing increased computational complexity. Therefore, in the control of the output voltage studied in this article, the dynamics of the current does not need to be taken into account, and by combining the complexity and accuracy of the modeling, the choice of the averaging model will be the most reasonable method.

In this article, the average model of the DAB converter is established in 'Modelling of a Dual Active Bridge Converter'. 'Controller Design' introduces the basics of the UDE scheme, which simplifies the design of the controller by transforming the complex nonlinear converter model into a simple linearized model through small-signal variations. This section integrates the small-signal model of the DAB with the UDE method, enabling the controller's output to be directly shifted in comparison to the small-signal control law of the UDE scheme. 'Experimental Studies' verifies the effectiveness of the scheme through a series of comparison experiments. 'Conclusions' summarizes the experimental results of this article. It improves the dynamic performance of the system and reduces the tracking error compared to the conventional control scheme.

## MODELLING OF A DUAL ACTIVE BRIDGE CONVERTER

The structure of the dual active bridge converter is shown in Fig. 1.

The topology consists of five parts: two H-bridges consisting of eight switching tubes on the primary and secondary sides, a high-frequency isolation transformer with a ratio of n:1; an input voltage $V_1$, a load resistor R, and an output filtering capacitor C. The voltage of the load on the output side is $V_2$. In the single phase shift (SPS) mode of operation, the diagonal switching tubes on both sides of the full bridge conduct and turn off at the same time, and the on and off time each accounts for 50% of the switching period T. The output waveforms of both sides of the full bridge are square waves, assuming that the converter's operating frequency is f, and the switching half-cycle of the drive signal is $T_s = 0.5/f = 0.5T$, and defining the shift ratio of the primary and secondary bridges as $D$ $(0 < D < 1)$, then the operating waveform of DAB at $t_0 = 0$, $t_2 = DT_s$, $t_3 = T_s$, $t_5 = T_s(1+D)$, $t_6 = T$ is shown in Fig. 2.

The inductor current expression at these four times is as follows:

$$i_L(t) = i_L(t_0) + \frac{V_1 + nV_2}{L}(t - t_0), t \in [t_0, t_2] \tag{1}$$

$$i_L(t) = i_L(t_2) + \frac{V_1 - nV_2}{L}(t - t_2), t \in [t_2, t_3] \tag{2}$$

$$i_L(t) = i_L(t_3) - \frac{V_1 + nV_2}{L}(t - t_3), t \in [t_3, t_5] \tag{3}$$

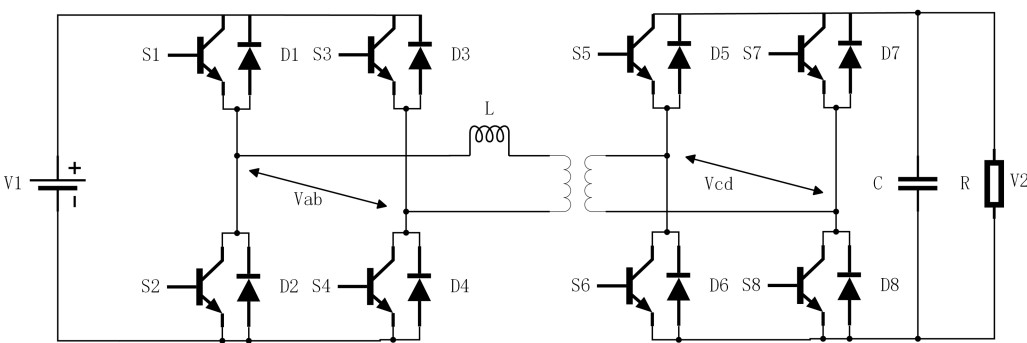

**Figure 1** **The structure of the dual active bridge converter.**

$$i_L(t) = i_L(t_5) + \frac{-V_1 + nV_2}{L}(t - t_5), t \in [t_5, t_6]. \tag{4}$$

From the above analysis of the modulation principle and the operating state, it can be seen that when the system is in a steady state, the following conditions exist:

$$i_L(t_0) = -i_L(t_3) \tag{5}$$

$$i_L(t_2) = -i_L(t_5.) \tag{6}$$

Combining the above conditions and the current Eqs. (1) to (6) for each mode, the current expressions for moments $t_0$ and $t_2$ can be found as follows:

$$i(t_0) = \frac{1}{4fL}[nV_2(1 - 2D) - V_1] \tag{7}$$

$$i(t_2) = \frac{1}{4fL}[nV_2 - V_1(1 - 2D)]. \tag{8}$$

The current in the secondary capacitor C varies periodically in four linear differential equations with an expression for each subinterval as follows

$$C\frac{dV_2}{dt} = -ni_L(t) - \frac{V_2}{R}, t \in [t_0, t_2] \tag{9}$$

$$C\frac{dV_2}{dt} = ni_L(t) - \frac{V_2}{R}, t \in [t_2, t_3] \tag{10}$$

$$C\frac{dV_2}{dt} = ni_L(t) - \frac{V_2}{R}, t \in [t_3, t_5] \tag{11}$$

$$C\frac{dV_2}{dt} = -ni_L(t) - \frac{V_2}{R}, t \in [t_5, t_6]. \tag{12}$$

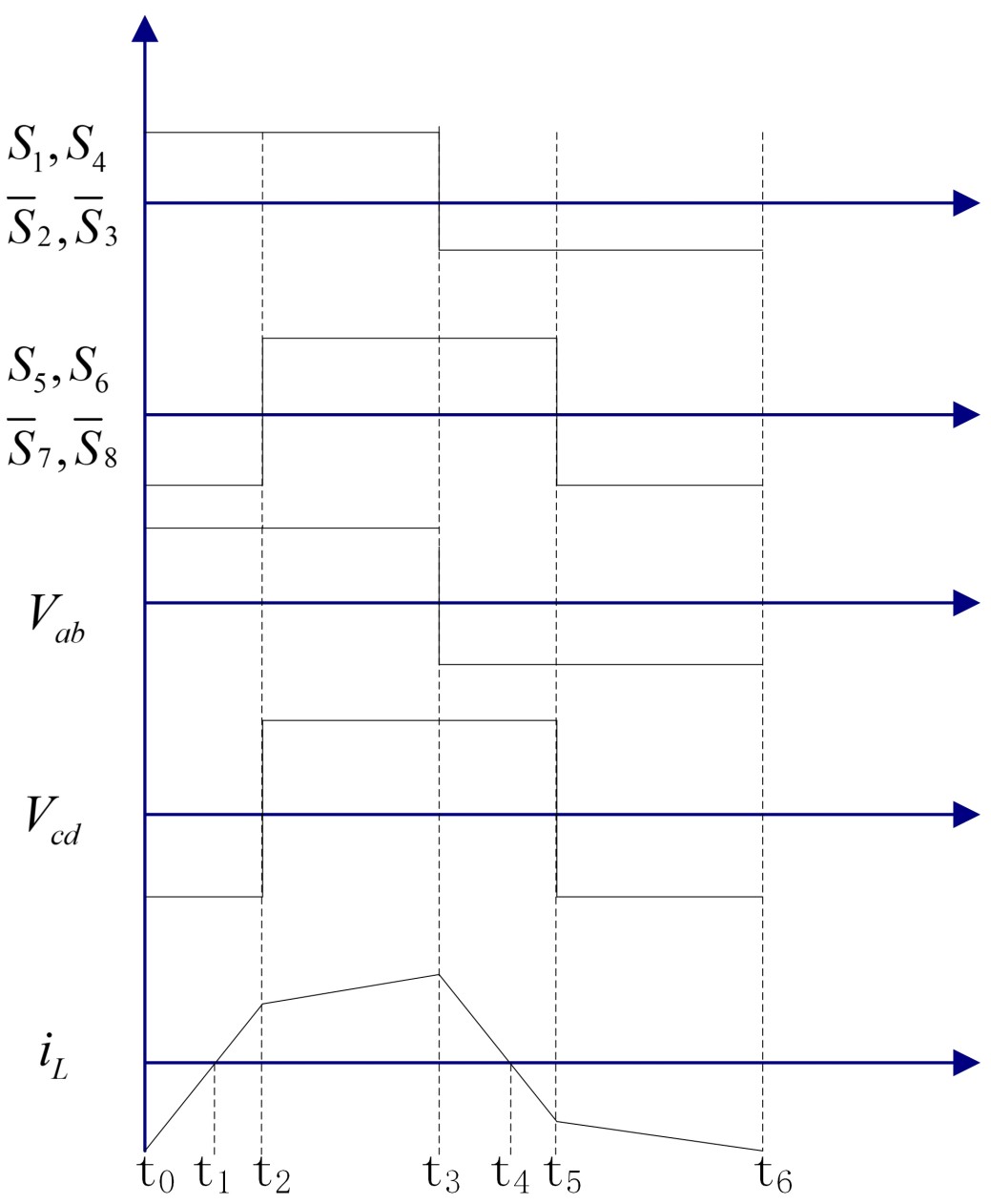

**Figure 2** Operating timing of the dual active bridge converter.

From Fig. 2, we can see that the DAB operating waveform is symmetric, then we only need to analyze its t∈[0, $T_s$] part, which can be obtained by combining Eqs. (1) to (4) in the above four parts:

$$C\frac{dV_2}{dt} = -\frac{V_1+V_2}{L}nt - \frac{n}{4fL}[V_2(1-2D)-V_1] - \frac{V_2}{R}, t\in[0,DT_s] \tag{13}$$

$$C\frac{dV_2}{dt} = \frac{V_1-V_2}{L}n(t-DT_s) + \frac{n}{4fL}[V_2-(1-2D)V_1] - \frac{V_2}{R}, t\in[DT_s,T_s]. \tag{14}$$

Averaging the above two Eqs. (13) and (14) over the period range of t $\in$[0, Ts] yields the average model of the DAB converter

$$\frac{dV_2}{dt} = -\frac{V_2}{RC} + \frac{nV_1}{2fLC}D(1-D).$$ (15)

When the system is in a steady state, $dV_2/dt$ is equal to 0, the relationship between the output voltage and the input shift ratio can be obtained as follows

$$V_2 = \frac{nV_1R}{2fl}D(1-D)$$ (16)

where it can be seen that the output voltage $V_2$ is related to the input voltage, load resistance, inductance, frequency, and shift ratio, and is independent of the transformer ratio n.

In the averaged equation, the input to the state equation is a quadratic function on the shift ratio D. Therefore, assuming that when the system is in the steady state, there exists a smaller transformation, the system model can be small-signalized, so that $V_2 = V_{2w} + v_2$, $D = D_w + d$, where $V_2w$, $D_w$ are the steady-state value of the system, $v_2$, $d$ are the small change of the system. combining the above conditions, the averaged Eq. (16) can be obtained

$$\dot{v}_2 = Av_2 + Bd$$ (17)

$$A = -\frac{1}{RC}, B = \frac{nV_1(1-2D_w)}{2fLC}.$$ (18)

At this point, we have obtained the state space expression for the model of the averaged small letter system, and also the expression for the transfer function of the shift ratio d to the output $v_2$. These expressions can be used for the design of the controller.

$$G(s) = \frac{v_2}{d} = \frac{nRV_1(1-2D_w)}{2fL(RCs+1)}.$$ (19)

## CONTROLLER DESIGN

### Introduction to UDE controllers

Consider a single-input single-output linear time-varying uncertainty system

$$\begin{cases} x^{\cdot}(t) = (A+\Delta A)x(t) + (B+\Delta B)u(t) + f(t) \\ y(t) = x_1 \end{cases}$$ (20)

where x(t) is the state vector of the system and x(t) = $[x_1, x_2, \ldots, x_n]^T$, A is the known state matrix of the system, B is the gain of the system's inputs, $\Delta$ A $\Delta$ B is the system's uncertainty term, which consists of the system component's parameter ingress and so on, u(t) is the system's inputs, and f(t) is the system's presence of external perturbations.

For the above system, a stable reference model needs to be chosen to satisfy the desired specifications of the closed-loop control system, such as the amount of overshoot of the system, the regulation time, *etc.* The reference model is assumed to be

$$\dot{x}_m(t) = A_m x_m(t) + B_m c(t)$$ (21)

where $x_m(t)$ is the state vector of the reference and $c(t)$ is the given signal to the reference model.

To achieve close tracking of the reference state $x_m(t)$, the tracking error is defined as

$$e(t) = x_m(t) - x(t) \tag{22}$$

and define the tracking error to satisfy

$$e^{\cdot}(t) = (A_m + K)e(t) \tag{23}$$

where K is the error feedback gain matrix and $A_m+K$ satisfies the Hurwitz matrix condition.

This is obtained by combining Eqs. (20) to (23) above

$$Ke(t) = A_m x(t) + B_m c(t) - Ax(t) - Bu(t) - \Delta Ax(t) - \Delta Bu(t) - f(t) \tag{24}$$

$$Bu(t) = A_m x(t) + B_m c(t) - Ax(t) - Ke(t) - u_d \tag{25}$$

where ud is the aggregate perturbation, including system uncertainty and external perturbations.

$$u_d = \Delta Ax(t) + f(t) + \Delta Bu(t) = x^{\cdot}(t) - Ax(t) - Bu(t). \tag{26}$$

The following conditions need to be satisfied when an exact solution to Eq. (26) exists

$$(I - BB^+)(A_m x(t) + B_m c(t) - Ax(t) - Ke(t) - u_d) = 0 \tag{27}$$

where $B^+ = (B^T B)^{-1} B^T$ is the pseudo-inverse of B and I is the unit matrix. Up to this point, the uncertainties and perturbations of the system can be observed as a known function of the system state and control signals. However, it cannot be used directly in the control law. Therefore the method of adding a filter is used to estimate the uncertainties and disturbances to construct the control law. Assuming that the filter $G_f(s)$ has unity gain over the full frequency range of the signal $u_d(t)$, $u_d(t)$ can be approximated as

$$\begin{aligned} u_{de}(t) \quad &= u_d * g_f(t) \\ &= (x^{\cdot}(t) - Ax(t) - Bu(t)) * g_f(t) \end{aligned} \tag{28}$$

where $^*$ is the convolution operator and $g_f(t)$ is the impulse response of $G_f(s)$.

Replace the set total perturbation $u_d(t)$ in Eq. (25) with its estimated value

$$Bu(t) = A_m x(t) + B_m c(t) - Ax(t) - Ke(t) - (x^{\cdot}(t) - Ax(t) - Bu(t)) * g_f(t). \tag{29}$$

Thus the UDE $-$based control law can be written as

$$\begin{aligned} u(t) = B^+ [-Ax(t) - L^{-1}\{\frac{sG_f(s)}{1 - G_f(s)}\} * x(t) + L^{-1}\{\frac{1}{1 - G_f(s)}\} \\ * (A_m x(t) + B_m c(t) - Ke(t))] \end{aligned} \tag{30}$$

where $L^{-1} \bullet$ is the Laplace inverse transform.

## Design of UDE controller for DAB converter

When there are uncertainty perturbations and external disturbances in the system, combined with Eq. (17), the state space equation of the DAB converter can be written as Eq. (20).

Based on the analysis in the previous chapters, it is known that the designed UDE control method should be based on a small signal model, so according to the following conditions:

$$V_2(t) = V_{2w}(t) + v_2(t) \tag{31}$$

$$D = D_w + d \tag{32}$$

$$V_{2m}(t) = V_{2wm}(t) + v_{2m}(t) \tag{33}$$

$$V_{ref} = V_{wref} + v_{ref} \tag{34}$$

$$e(t) = e_w(t) + \Delta e(t) \tag{35}$$

where $V_{2w}$, $D_w$, $V_{2wm}$, $V_{ref}$, $e_w$ are the output voltage at the steady state point, the shift ratio, the output voltage of the reference system, the expected value of the voltage, and the tracking error; $v_2$, d, $v_{2wm}$, $v_{ref}$, and $\Delta e$ are based on the small variations of the respective signals at the steady state point, respectively. Combined with the principle of the UDE from the previous section:

$$\dot{V}_{2wm}(t) = -\alpha V_{2wm}(t) + \alpha v_{ref} \tag{36}$$

$$\Delta e(t) = v_{2m}(t) - v_2(t) \tag{37}$$

$$\Delta \dot{e}(t) = (-\alpha - K)\Delta e(t) \tag{38}$$

$$Bd(t) = -\alpha v_2(t) + \alpha v_{ref}(t) - Av_2(t) + K\Delta e(t) - u_d \tag{39}$$

$$u_d = \Delta Av_2(t) + \Delta Bu(t) + f(t) = v_2'(t) - Av_2(t) - Bd(t) \tag{40}$$

$$u_{de} = (v_2'(t) - Av_2(t) - Bd(t)) * g_f \tag{41}$$

where the reference system needs to satisfy stability, then $\alpha > 0$. $\alpha + k$ needs to satisfy the Hurwitz matrix condition.

Thus the control law for the DAB input shift ratio d in the small signal model can be obtained as

$$d(t) = B^+[-Av_2(t) - L^{-1}(\frac{sG_f(s)}{1 - G_f(s)}) * v_2(t) + L^{-1}(\frac{1}{1 - G_f(s)})$$
$$* (-\alpha v_2(t) + \alpha v_{ref}(t) + K\Delta e(t))].$$

Bringing Eqs. (37) to (41) into (20) yields

$$\dot{v}_2 = -\alpha v_2(t) + \alpha v_{ref}(t) + K\Delta e(t) - u_d * g_f + u_d. \tag{43}$$

The expression for the system output concerning the desired input concerning the aggregate perturbation can be obtained by taking the Rasch transform of the above equation and simplifying it

$$v(s) = \alpha(s + \alpha)^{-1}v_{ref}(s) + (s - (-\alpha - K))^{-1}(1 - G_f(s))U_d(s) \tag{44}$$

in which

$$H_m = \alpha(s + \alpha)^{-1} \tag{45}$$

$$H_d(s) = (sI - (-\alpha - K))^{-1}(1 - G_f(s)) \tag{46}$$

$$H_k(s) = (sI - (-\alpha - K))^{-1} \tag{47}$$

$$H_f(s) = 1 - G_f(s) \tag{48}$$

where $H_m$ is the same as the reference model $V_{ref}$ to $V_{2wm}$ transfer function given in Eq. (36), and the transfer function is independent of the error feedback gain k. The second half of Eq. (44) represents the system error after the set total perturbation is attenuated through $H_d$. It can be observed that the UDE controller essentially possesses a structure with two degrees of freedom. Here, the reference model defines the desired system response point, while the perturbations are determined by the error feedback gain k in conjunction with the filter.

The front and back parts of Eq. (44) need to be stabilized separately according to the superposition principle to achieve a stable output. When the reference system is chosen, the transfer function of the reference system is always stabilized when $\alpha > 0$. Also, since the desired output $V_{ref}$ of the system is always bounded, the stabilization of the first half of Eq. (44) can be achieved. For the stabilization of $H_d$, it is required that $-\alpha - K < 0$, and additionally, it is required that the filter is a strictly stable filter and the aggregate perturbations are also bounded so that ultimately the system can be realized with bounded inputs and bounded outputs.

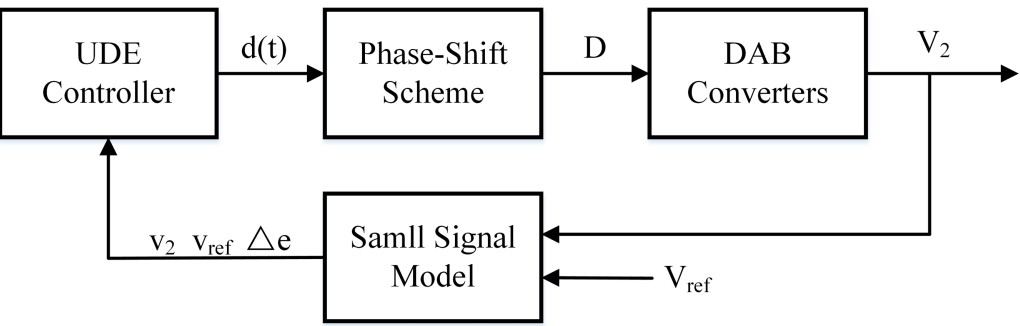

**Figure 3** **System block diagram of DAB converter based on UDE controller.**

So far, in the UDE controller, the reference system, error feedback gain K, filter $G_f(s)$, and the block diagram of the UDE-controlled DAB converter are required to be designed as shown in Fig. 3.

Assume that the general form of the filter is as follows:

$$G_f(s) = \frac{1}{\tau s + 1} = \frac{\beta}{s + \beta}.$$ (49)

The tracking error of the system can be written as

$$E(s) = -H_d(s)U_d(s).$$ (50)

Combining Eqs. (47) to (50), the $H_d$, $H_k$, and $H_f$ amplitude–frequency characteristic curves can be drawn as shown in Fig. 4. When $H_k(s)$ is kept constant and the bandwidth of filter $G_f(s)$ increases, *i.e.,* when $\tau$ decreases and $\beta$ increases, the curve Hf(s) will move downward in the low-frequency band, forcing the curve $H_d(s)$ to move downward in the low-frequency band as well, which results in a more pronounced attenuation effect on the low-frequency band of the aggregate perturbation. Therefore, the larger the bandwidth of the filter $G_f(s)$, the stronger the anti-disturbance performance and the smaller the tracking error of the UDE controller.

## EXPERIMENTAL STUDIES

In order to verify the effectiveness of applying the UDE controller to the DAB converter, a simulation model of the system was built in the simulation software by applying the parameter data in Table 1. From the previous section, it can be seen that the model is based on a small signal, so it is necessary to select a stable point, according to Eq. (16), selected at an input voltage of 400 V and an output voltage of 400 V. The other parameters are shown in Table 1, which gives the value of the shift ratio at the steady state point as 0.052786. To verify the superiority of the UDE controller, a PI controller was also designed($PI(s) = k_p + k_i/s$),$k_p = 7.143*10^{-4}$,$ki = 6.525*10^{-2}$. The closed-loop bandwidth of the control system is the same as the reference system bandwidth of the UDE controller and the system is critically damped under PI control. The reference system $\alpha$ for UDE control is 300, the error feedback gain k is 0 (*Zhong & Rees, 2004*), and $\beta$ in the filter is 600, and the following simulations are performed.

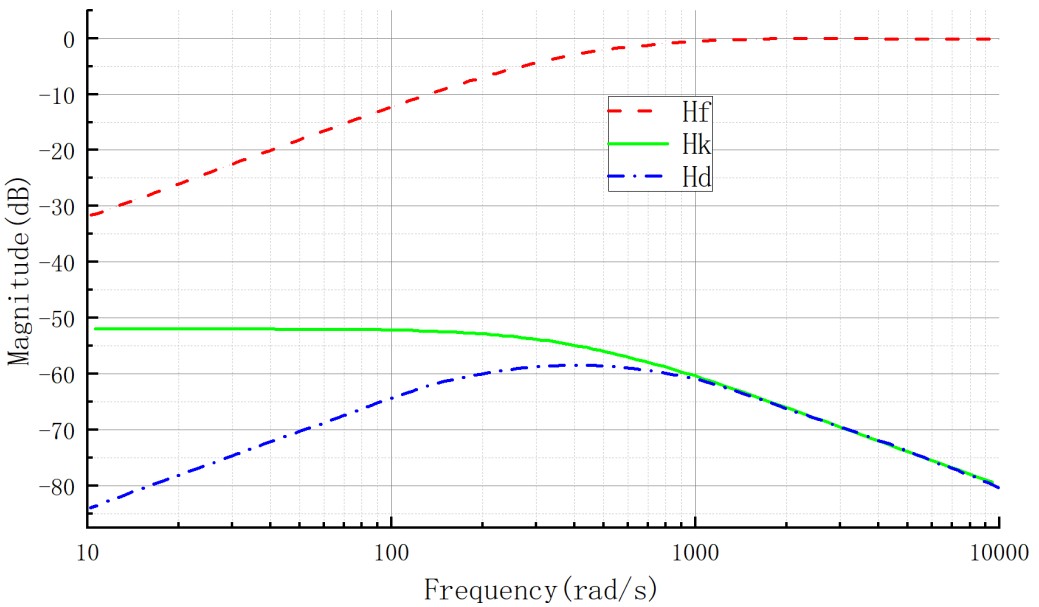

**Figure 4** $H_d$, $H_k$, $H_f$ amplitude–frequency characteristic curve.

**Table 1** DAB circuit parameters.

| Parameter | Value | Parameter | Value |
|---|---|---|---|
| Input voltage V1 | 400 V | Reference V2 | 420 V |
| Turns ratio | n | Switching frequency | 20 kHz |
| Leakage inductance L | 125 μH | Output capacitor C | 400 μF |
| Load resistor R | 50 Ω | | |

Figure 5 shows the transient response curve under the control system of the PI controller and UDE controller when the desired voltage changes. In the figure, UDE-V2 represents the curve of the system output voltage under the control of the UDE controller. PI-V2 represents the curve of the system output voltage under the control of the PI controller. Vref represents the desired value of the system output voltage. UDE-e represents the error of the system output voltage under the control of the UDE controller. PI-e represents the error of the system output voltage under the control of the UDE controller. V1 represents the system source-side voltage. R represents the load resistance of the system. Before 0.1 s, the output voltage of the system is stabilized at 400 V, and the desired voltage is changed from 400 V to 370 V during 0.1 s, other conditions remain unchanged, and the output voltage of the UDE control system reaches the desired value of the output voltage after 0.02 s, and the output voltage of the control system of PI controller reaches the desired value of the output voltage after 0.04 s. The output voltage of the UDE control system reaches the desired value of output voltage after 0.02s, and that of the PI controller control system reaches the desired value of output voltage after 0.04s. It can be seen that the UDE controller is able to track the desired value of the voltage, and compared with the PI controller, the

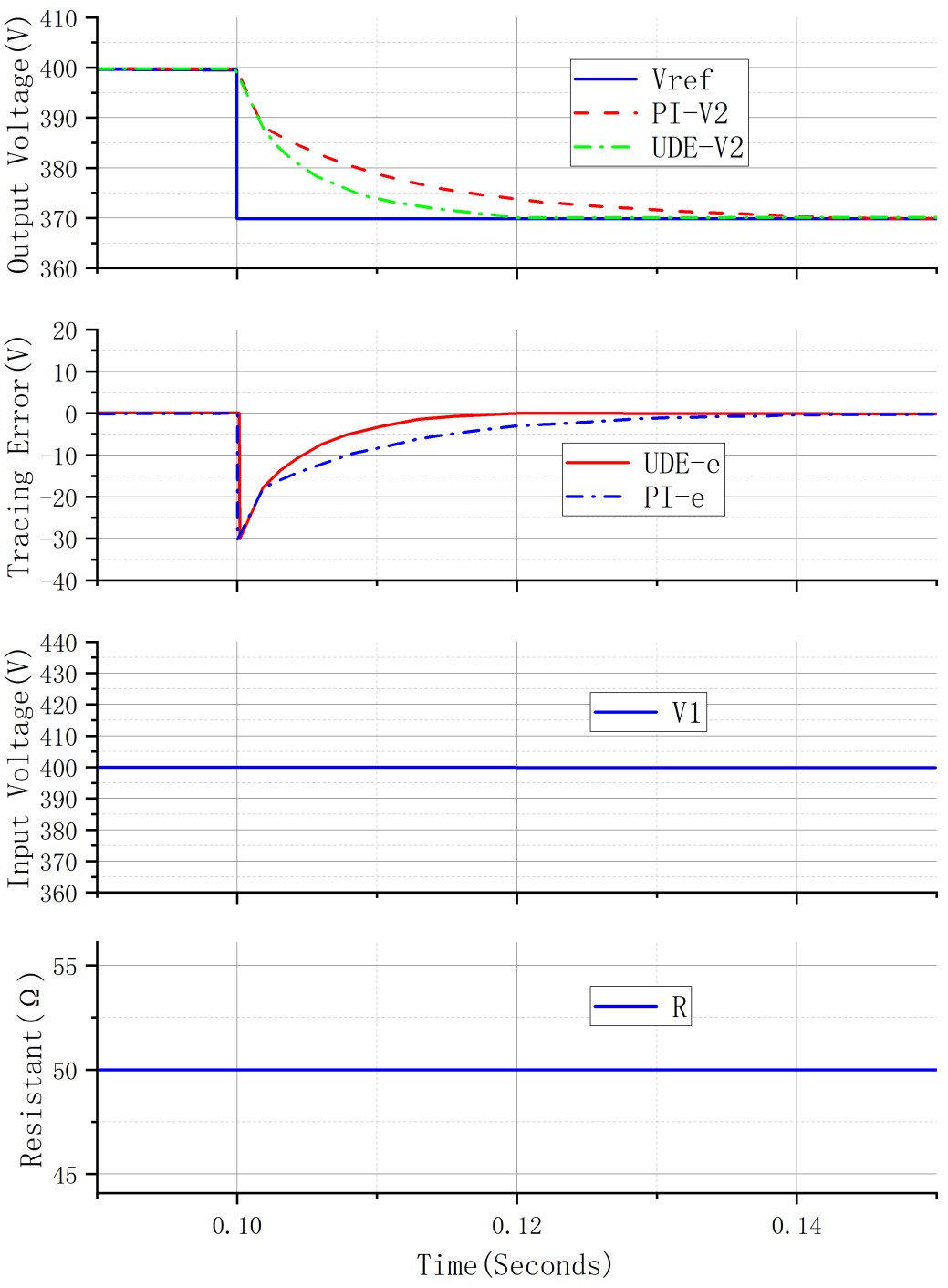

**Figure 5** **V**$_{ref}$ **step change system response curve.**

UDE controller shows better dynamic performance, the waveform of the output voltage is smoother, and the time to reach the steady state is shorter.

Figure 6 shows the performance of two different controllers in controlling the output voltage during the step change of input voltage, the input voltage changes from 400 V

to 500 V in 0.3 s, and the output voltage of the UDE control system reaches the desired value of the output voltage after 0.02 s, and the fluctuation value of the voltage is 6 V, and the output voltage of the PI controller control system reaches the desired value of the output voltage after 0.05 s, and the fluctuation value of the voltage is 14 V. This shows that the UDE controller has better control performance than the PI controller when the input voltage changes drastically.

Figure 7 shows the response curve of the system when the load resistance changes stepwise, the load resistance changes from 50 Ω to 75 Ω in 0.5s, the output voltage of the UDE control system reaches the desired value of the output voltage after 0.02s, and the fluctuation value of the voltage is 5V, and the output voltage of the PI controller control system reaches the desired value of the output voltage after 0.05s, and the fluctuation value of the voltage is 13V. Compared with the PI controller, the UDE can converge to the desired value of the voltage faster and the voltage fluctuation is smaller. Compared with the PI controller, the UDE can converge to the desired value of the voltage faster and the voltage fluctuation is smaller.

In addition to further verify the performance of the designed controller under continuous disturbance, the load resistance is set to R = 50+20sin(100t). From Fig. 8, it can be seen that the output voltage fluctuation of the UDE controller control under continuous disturbance is smaller, $400 \pm 5$V, and the output voltage curve is closer to the expected value of 400V, while the output voltage fluctuation of the PI controller control is $400 \pm 13$V.

In conclusion, the UDE control has better performance in the face of system uncertainties and disturbances.

## CONCLUSIONS

In this article, a UDE-based robust voltage control scheme for DAB converters is proposed, which is used to improve the tracking performance and the suppression of the system against internal and external disturbances and uncertainties. In order to simplify the design of the controller and improve the compatibility of the controller, a generalized DAB converter small-signal model based on converter shift comparison is proposed. Then, using the proposed universal DAB model, a UDE-based voltage controller is proposed to analyze the conditions for the UDE controller to achieve the stability of the output voltage of the DAB converter. The parameters that need to be designed are the closed-loop bandwidth of the reference system $\alpha$, the error feedback gain K, and the filter bandwidth $\beta$. When the stability is satisfied, controller design is conducted under conditions encompassing step changes in input voltage, step changes in load resistance, continuous variations in load resistance, and so on. Under the conditions of input voltage step change, load resistance step change, load resistance continuous change, *etc.*, the system output response is simulated and experimented, and the results verify the effectiveness and superiority of the program.

In future work, the UDE method can also be used to analyze the non-matching uncertainty of the system, however, the perturbation of the non-matching uncertainty is not reflected in our model, so it is necessary to consider more suitable modeling methods. In addition, it is necessary to continue optimizing the reference system and parameter

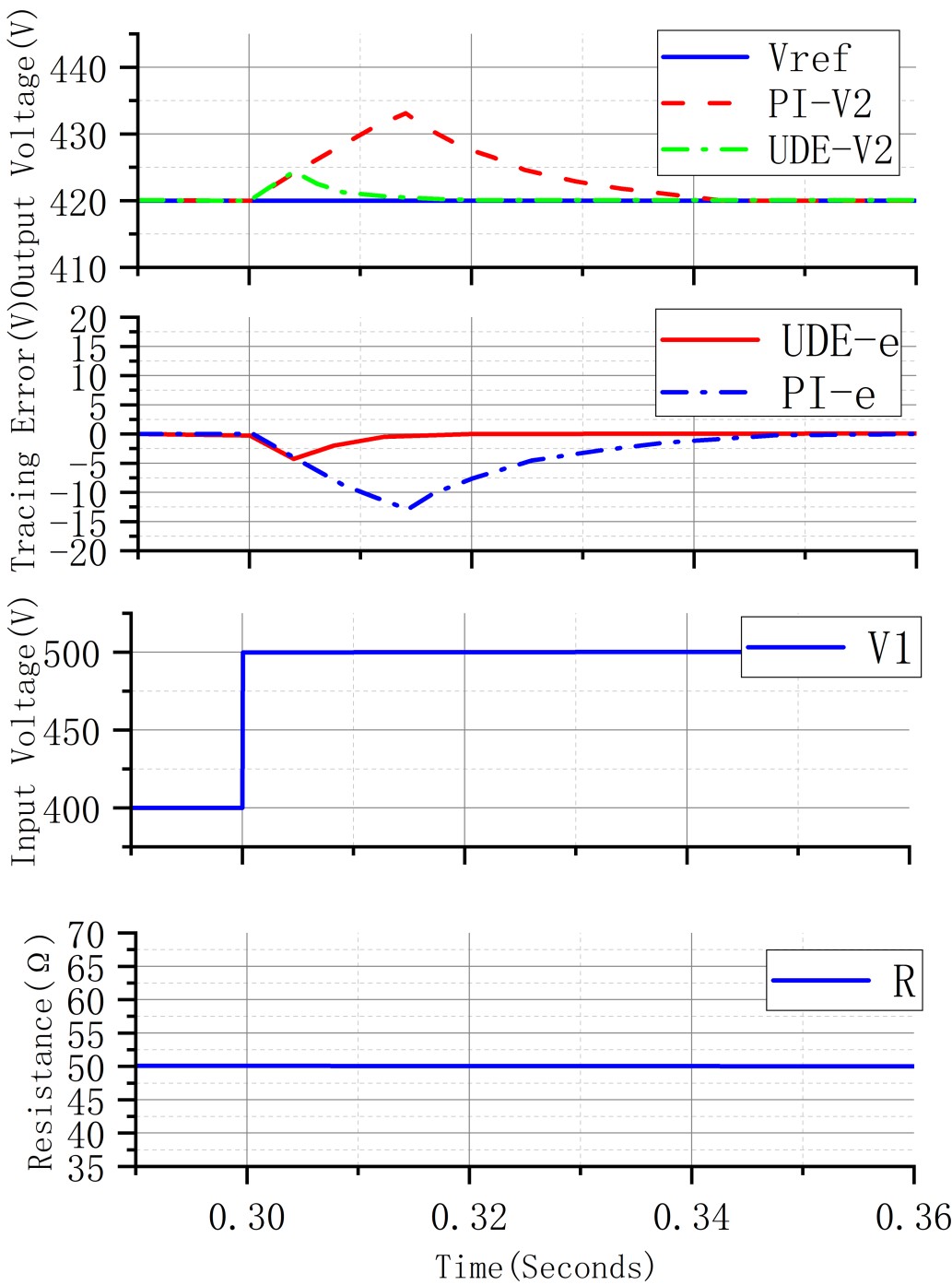

Figure 6  Input voltage step change system response curve.

selection. In this article, the simulation was only carried out in the simulation software, and the hardware platform was not built, which is a point that needs to be continued research in the future. In this way, tracking performance and dynamic response of the system can be better achieved.

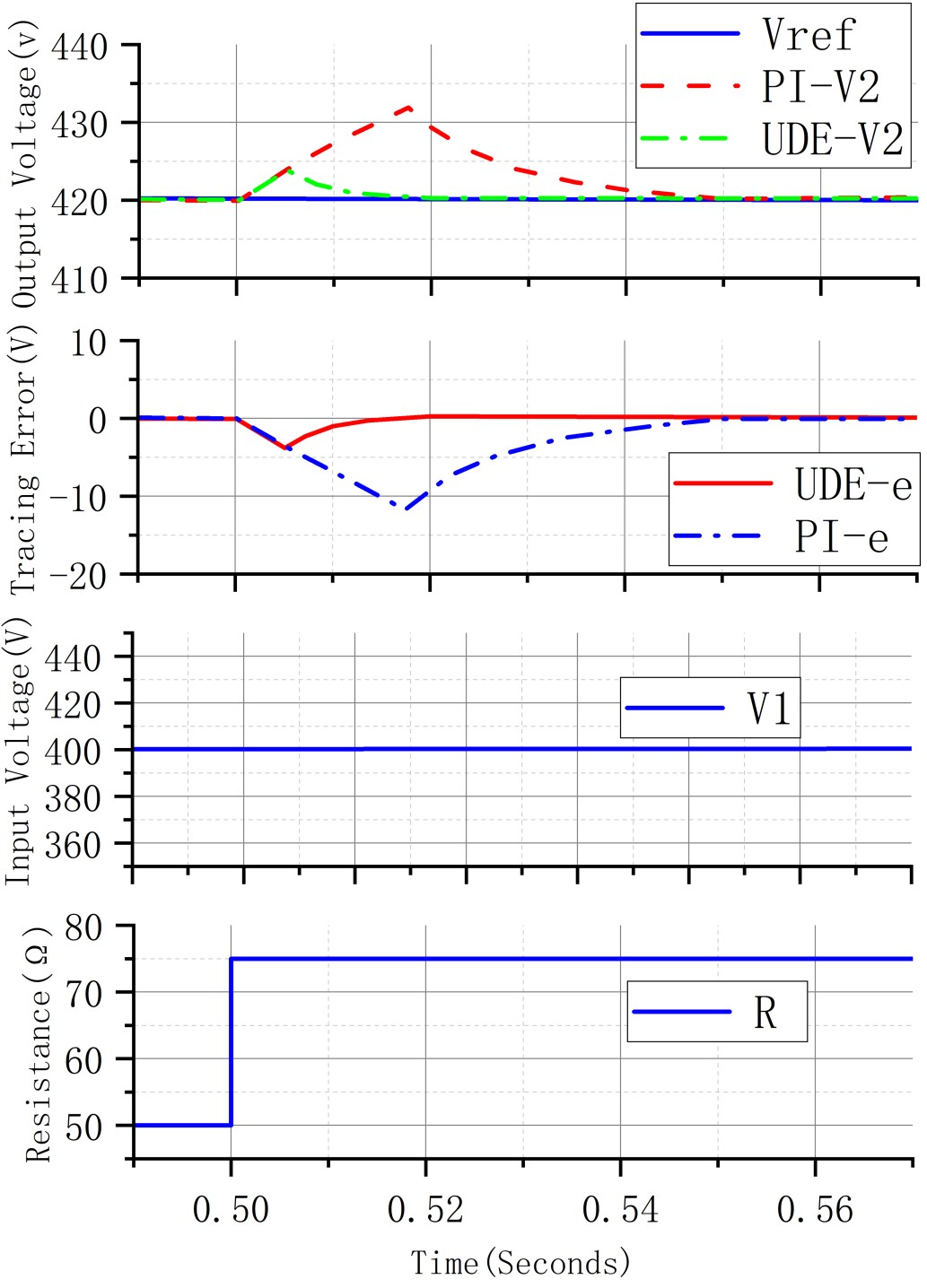

**Figure 7** System response curve for step change in load resistance.

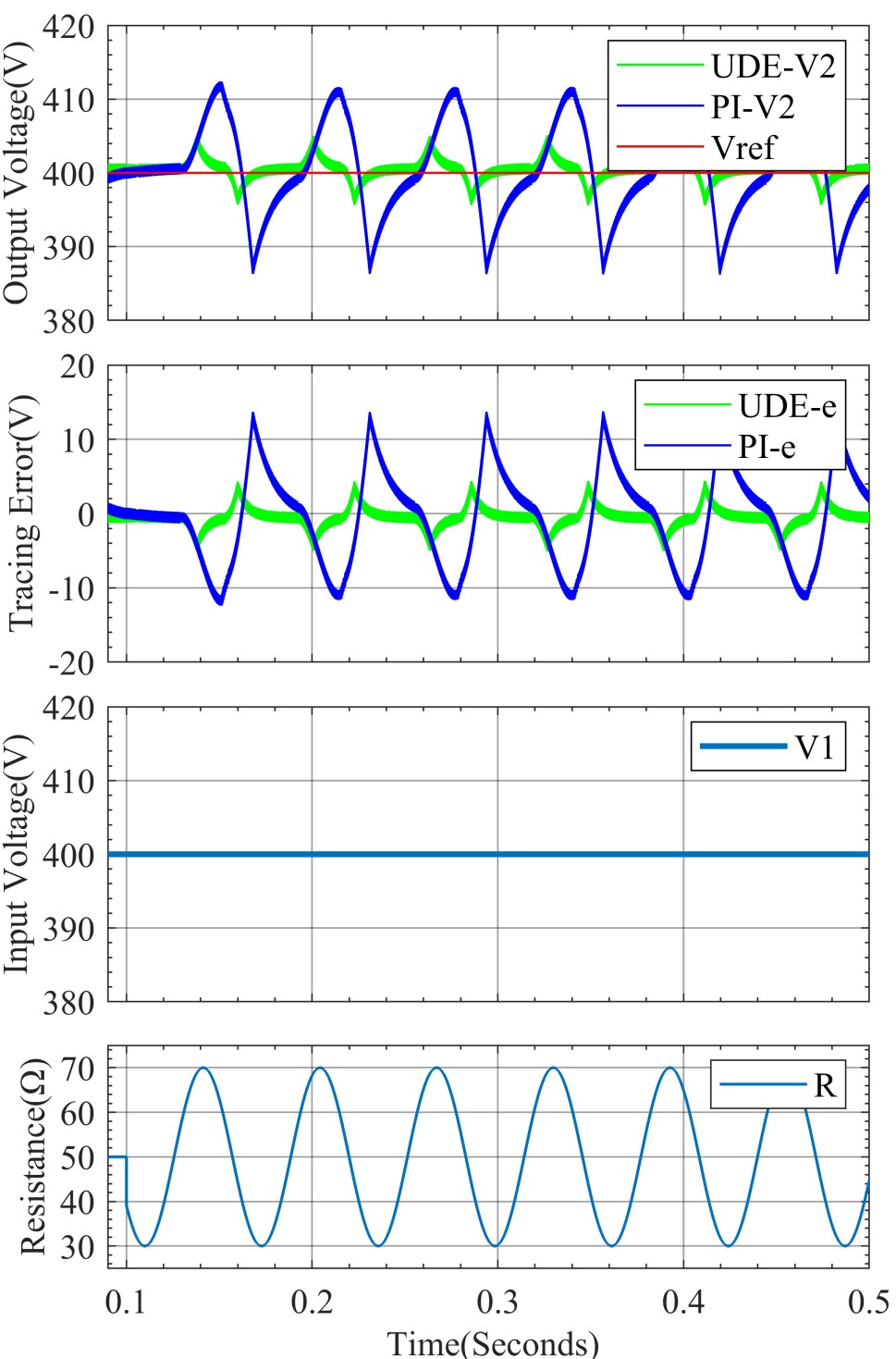

**Figure 8  System response curve for continuous change in load resistance.**

### Funding
The authors received no funding for this work.

### Competing Interests
The authors declare there are no competing interests.

### Author Contributions

- Haijun Tian conceived and designed the experiments, performed the experiments, analyzed the data, performed the computation work, authored or reviewed drafts of the article, and approved the final draft.
- Zheng Zhou conceived and designed the experiments, analyzed the data, performed the computation work, prepared figures and/or tables, authored or reviewed drafts of the article, and approved the final draft.
- Yuanshuai Liu conceived and designed the experiments, performed the experiments, analyzed the data, performed the computation work, authored or reviewed drafts of the article, and approved the final draft.
- Yuepeng Zhang conceived and designed the experiments, prepared figures and/or tables, and approved the final draft.
- Mingsuo Yang conceived and designed the experiments, prepared figures and/or tables, and approved the final draft.

### Data Availability
The code and raw data are available in the Supplemental Files.

### Supplemental Information
Supplemental information for this article can be found online at http://dx.doi.org/10.7717/peerj-cs.2175#supplemental-information.

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
