# Peer review of "Output voltage control of DAB converters based on uncertainty and disturbance estimation"

_PeerJ Computer Science, doi:10.7717/peerj-cs.2175_

## Round 0.1 · original submission · Major Revisions

According to the reviewers, the manuscript must be revised accordingly.

**Language Note:** The review process has identified that the English language must be improved. PeerJ can provide language editing services - please contact us at [email protected] for pricing (be sure to provide your manuscript number and title). Alternatively, you should make your own arrangements to improve the language quality and provide details in your response letter. – PeerJ Staff

Reviewer 1 ·

Basic reporting

This manuscript endeavors to provide a new method to control the output voltage of DAB converters. Below are some areas of improvement and suggestions to enhance the clarity, coherence, and depth of the manuscript:
1. The introduction could benefit from a more detailed exposition of the research problem, the existing gap in the literature presented by the other authors (Wu et al., 2020 and/or others), and a clear explanation of your contribution to improving this area. Clearly articulate the purpose, scope, and objectives of the study.

2. There are a lot of typos. I advise a thoughtful revision of the document. Some typos are listed below:
• Line 32: PI not defined;
• Line 46: SMC not defined;
• Line 58: UDE not defined;
• Fig 1 should have V2 (in the load) clearly defined because you use it along the manuscript;
• Line 73: SPS not defined;
• Line 78: Typo in t3=Ts; should be t2;
• Figure 2: Vad and Ved are not defined in the text, neither represented in Figure 1;
• Eq 20 is missing deltaA and deltaB
• Typo in eq 21 (the m subscript)?;
• Eq 31 is equal to eq. 20; You only need one!
• Line 112, you have Section (without the reference to each);
• Typo in eq 37 (the 2wm subscript)?;
• Line 152 k=0, and there’s a reference. Are they misplaced or just lacking a space?

3. Proper citation is lacking. Some more recent studies could be added to this manuscript (only 1 citation from 2023, 0 from 2022, 1 from 2021, 2 from 2020). You mention in the introduction: “Literature (Wu et al., 2020) applies the UDE control method for a constant power load of a DAB converter, and for the first time, the output impedance of a UDE-controlled DAB converter is modeled and analyzed for stability”. I would expect your proper mention of the recent additions in this field.

4. The paper is written at a proficient level of English.

Experimental design

The study technique session would benefit from enhancements in the explanation of methodology in the sense that you present a great number of equations, and there are many particularities and assumptions that you could consider further enhancing. Have a thoughtful revision of the Controller Design and Experimental Studies sections.
1. As an example (but there are others), how did you accomplish to obtain the PI constants presented in line 149? Why did you use the testbed for a 20kHz frequency? And so on…

2. Figures 5 to 7 have UDE-v2 and PI-v2. Is it version 2 of something, or is it V2 (V load)?

Which parts of the methodology are from your development, and which are based on other studies? Because no references to other studies (if that’s the case, of course) are mentioned.

Validity of the findings

1. The major problem with this manuscript is the lack of comparison with other works in literature. The lack of a comparative basis with the most recent works in the scientific area does not allow us to clarify where this contribution stands in relation to the others. It is essential to the quality of this manuscript that this work is added to evaluate its impact.

2. The Conclusions section could delve deeper into how the findings address the problem and how they contribute to the evolution of this scientific area. Discussing the implications and applications of the results, alongside suggesting directions for future research, will enrich the manuscript. Also, acknowledging the limitations of the proposed approach in more depth could provide a more comprehensive understanding.

3. Although the work now presents a simulation with a model, it is still difficult to understand whether this model has real applicability. Were real tests carried out in the field or only simulations? What is the evidence that proves the added value of this approach to the field?

Reviewer 2 ·

Basic reporting

In this manuscript, the authors propose a robust control scheme for the output voltage based on uncertainty and disturbance estimation. In this paper, an averaged small-signal model of a dual active bridge converter is first established, outlining the basic principles and operational mechanisms, thereby simplifying the design of the controller. Next, the authors introduce the basic principles of the uncertainty and disturbance estimator. They apply the small-signal model of the dual active bridge converter to the uncertainty and disturbance estimator to reduce the error of the output voltage by utilizing the controller to directly control the shift ratio. Finally, the authors discuss the application and effectiveness of the uncertainty and disturbance estimator in the simulation and control of dual active bridge converters. They conduct a series of experimental comparative studies, indicating that this scheme has significant advantages in suppressing system uncertainty and disturbance.
The paper is interesting, and the results seem to be valid. Moreover, the presentation of the article is also well done. However, there are some major concerns:
Firstly, why did the authors establish an averaged small-signal model of a dual active bridge converter in terms of the basic principles and operational mechanisms, simplifying the controller's design compared to other models? What is the need for this establishment?
The introduction requires significant improvements, such as providing background information, stating the problem clearly, outlining the actual contributions, and including some results that would be better suited for inclusion in the abstract.
Figure 1 needs to be represented with better pixel quality.
Symbols need to be double-checked, as some are repeated for various terminologies.
There is a need for a table for symbols with their notations and descriptions.
A related work section should be included to distinguish the work from recent benchmarks. Are there any benchmarks that should be referenced and compared with the proposed study?

Experimental design

Included in basic reporting.

Validity of the findings

Included in basic reporting.

---

## Round 0.2 · accepted · Accept

Based on the reviewer comments, the manuscript can be accepted.

Reviewer 1 ·

Basic reporting

NA

Experimental design

NA

Validity of the findings

NA

Additional comments

English and typos improved from last version;
In general, all the previous concerns were addressed;

Reviewer 2 ·

Basic reporting

I have carefully examined the rebuttal and revised manuscript, and found that authors have carefully tackled all comments. I am happy to recommend the acceptance of the article.

Experimental design

No comments

Validity of the findings

No comments